# CDNet: Causal Inference inspired Diversified Aggregation Convolution for Pathology Image Segmentation

**Dawei Fan**[1]                                  DAWEI.FAN@FAFU.EDU.CN
**Yifan Gao**[1]                                  121193006@FAFU.EDU.CN
**Jiaming Yu**[1]                                  17689418024@163.COM
**Changcai Yang**[1,2]                            CHANGCAIYANG@GMAIL.COM
**Riqing Chen**[1,2]                              RIQING.CHEN@FAFU.EDU.CN
**Lifang Wei**[1,2,*]                             WEILIFANG0028@FAFU.EDU.CN

[1]*College of Computer and Information Sciences*
*Fujian Agriculture and Forestry University*
*Fuzhou 350002, China.*
[2]*College of Future Technology*
*Fujian Agriculture and Forestry University*

## Abstract

Deep learning models have shown promising performance for Nuclei segmentation in the field of pathology image analysis. However, training a robust model from multiple domains remains a great challenge for Nuclei segmentation. Additionally, the shortcomings of background noise, highly overlapping between Nuclei, and blurred edges often lead to poor performance. To address these challenges, we propose a novel framework termed CDNet, which combines Causal Inference Module (CIM) with Diversified Aggregation Convolution (DAC) techniques. The DAC module is designed which incorporates diverse downsampling features through a simple, parameter-free attention module (SimAM), aiming to overcome the problems of edge blurring. Furthermore, we introduce CIM to leverage sample weighting by directly removing the spurious correlations between features for every input sample and concentrating more on the correlation between features and labels. Extensive experiments on the MoNuSeg and GLySAC datasets yielded promising results, with mean intersection over union (mIoU) and Dice similarity coefficient (DSC) scores increasing by 3.59% and 2.61%, and 2.71% and 2.04%, respectively, outperforming other state-of-the-art methods.

**Keywords:** Causal inference, Feature aggregation, Nuclei segmentation, Pathology image

## 1 Introduction

[1] Over the past decade, deep learning has reached promising results for Nuclei segmen-tation in the field of pathology image analysis (Ronneberger et al., 2015; Zhou et al., 2018; Oktay et al., 2018; Chen et al., 2021; Cao et al., 2022). For instances, the U-Net pro-posed by (Ronneberger et al., 2015) and its improved version, U-Net++ by (Zhou et al., 2018), with their unique encoder-decoder structure and skip connections, effectively integrate low-level to high-level semantic information of images, which greatly promotes the development of Nuclei segmentation technology. Moreover, the introduction of the Transformer (Vaswani et al., 2017) architecture, with its unique attention mechanism (Schlemper et al., 2019), has optimized the processing of complex medical images, captured the intricate spatial relation-

---

1. * Corresponding author

ships and feature hierarchies within images. Additionally, TransUNet (Chen et al., 2021) combines the image encoding capabilities of Convolutional Neural Network (CNN) with the deep contextual understanding of visual Transformers (Dosovitskiy et al., 2020), bringing new perspectives and powerful potential to Nuclei segmentation technology. The application of classical medical image segmentation models to pathological sections, in particular those containing cell nuclei, is a challenging endeavour. In 2019, (Graham et al., 2019) introduced HoVer-Net, which employs horizontal and vertical distance maps to effectively separate overlapping cell nuclei. In 2022, (Wang et al., 2022) developed UCTransNet, an optimised U-Net architecture that improves segmentation by narrowing the semantic gap and exploiting multi-scale features. However, existing models encounter difficulties in the presence of background noise, overlapping nuclei, and blurred edges (Fig. 1), which negatively impact segmentation accuracy and clarity. These models are unable to distinguish between closely spaced nuclei and their surrounding tissue with sufficient accuracy. Furthermore, variations in lighting, resolution, and staining techniques across different hospitals or devices present challenges for the application of deep learning models in clinical practice.

To tackle these challenges, we propose the use of CDNet, which combines the Causal Inference Module (CIM) with Diversified Aggregation Convolution (DAC). The DAC module employs a variety of downsampling techniques through SimAM (Yang et al., 2021) to mitigate edge blurring, thereby enhancing the precision and clarity of nuclei segmentation. The CIM addresses the issue of performance decline resulting from variations in data distribution across hospitals and devices. It does so by leveraging sample weighting to remove spurious correlations and focus on meaningful feature-label relationships, as proposed by (Zhang et al., 2021). The following key contributions are made: (1) The DAC module was designed for feature fusion with the objective of optimising semantic information and reducing edge blurring issues. (2) The introduction of CIM with a causal inference strategy enables the dynamic adjustment of sample weights, thereby enhancing nuclei feature recognition and reducing spurious correlations. A comprehensive series of experiments has demonstrated that CDNet is a superior method to other state-of-the-art techniques, offering new insights for addressing domain shift in pathology image analysis and improving segmentation accuracy.

## 2 Method

In this section, we first introduce an overview of our CDNet. Then, we detail each part of the CDNet.

### 2.1 Overview

A graphical representation of the CDNet overview is provided in Fig. 2. The framework commences with an input micrograph image, which is subjected to five layers of CNN and multiscale block convolutional neural network (MBConv) (Tan and Le, 2019) downsampling, thereby creating multilevel feature maps. In the subsequent stage, the feature maps generated by each downsampling stage are combined using a dynamic attention mechanism (DAC). The Transformer enhances the initial CNN layer's feature maps, thereby producing a more integrated set. CIM learns the sample weights and computes the weighted loss for the final feature maps. The aforementioned maps are then fused using skip connections and

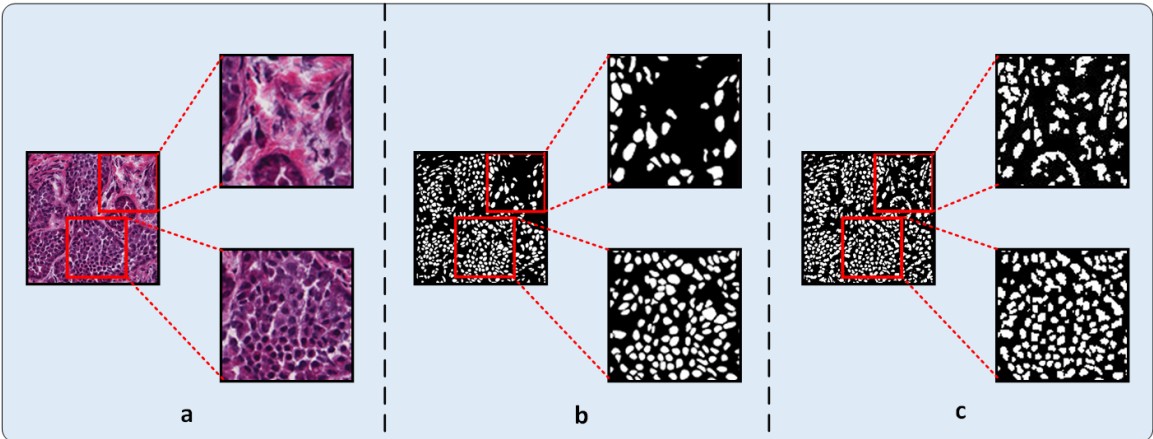

Figure 1: Nuclei segmentation problem illustration. (a) denotes the micrographs image (b) denotes the ground truth and (c) denotes the predicted results obtained from Attention U-Net.

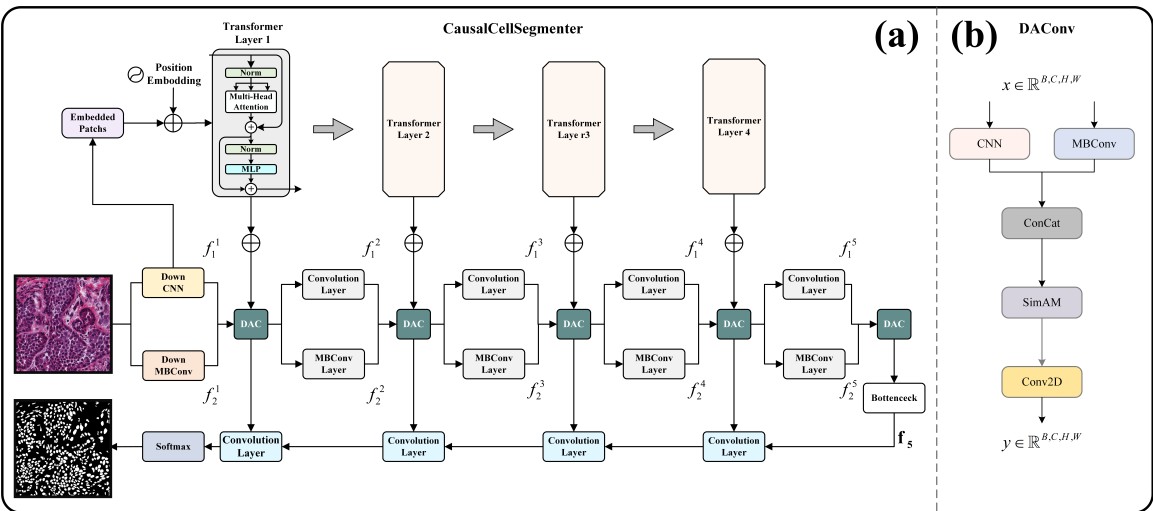

Figure 2: The overview of CDNet (a) with DAC module (b).

a convolutional decoder, in a manner analogous to that observed in U-Net. Ultimately, the softmax function is responsible for generating the nuclei segmentation map.

## 2.2 Diversified Aggregation Convolution (DAC)

To enhance the precision of Nuclei segmentation, particularly in addressing segmentation challenges arising from issues such as blurred edges and background noise, we introduce an effective Diversified Aggregation Convolution (DAC) module. The DAC is specifically designed to integrate multi-scale feature maps (Cui et al., 2016) with rich semantic information. It optimizes the feature extraction process using CNN and MBConv and focuses

on regions of interest by a simple, parameter-free attention module (SimAM), achieving precise discrimination between nucleus and non-nucleus.

As demonstrated in Fig. 2(b), the input data undergoes a downsampling $f_1^i$ and $f_2^i$ after using CNN and MBConv, where $f_1^i$ represents the i-th layer downsampling using CNN and $f_2^i$ represent the corresponding layer using MBConv. Afterwards, we concatenate $f_1^i$ and $f_2^i$ by learned weights $k_1^i$ and $k_2^i$. The obtained features are then fed into SimAM, which aims to accurately evaluate and determine the most important features of semantic information in the Nuclei segmentation. Subsequently, these features from SimAM are downsampled using two-dimensional convolution (Conv2D). Hence, the DAC can be represented as follows:

$$\mathbf{f_i} = \text{Conv2D}(\text{SimAM}([k_1^i f_1^i, k_2^i f_2^i]))\tag{1}$$

Where $[\bullet]$ denotes the CONCAT operation and $\mathbf{f_i}$ denotes i-th layer feature maps after DAC. We initialize $k_1^i$ and $k_2^i$ as 1.0.

### 2.3 Causal Inference Module (CIM)

To address data heterogeneity problem, we follow the StableNet (Zhang et al., 2021) framework and design a causal inference module (CIM). We obtain the feature maps $\mathbf{f_5}$ from the 5-th layer DAC as shown in Fig. 2. To remove spurious correlations between features, we leverage Random Fourier Features (RFF) extractor and sample weighting for segmentation task. Specifically, we use A and B represent the feature variable in the feature map from $\mathbf{f_5}$. As Frobenius norm of the partial cross-covariance matrix $\|\sum_{AB}\|_F^2$ tends to zero, the two variables A and B are independent (Zhang et al., 2021). Hence, the partial cross-covariance matrix be:

$$\Sigma_{AB} = \frac{1}{n-1} \sum_{i=1}^{n} \left[ \left( u(A_i) - \frac{1}{n} \sum_{j=1}^{n} u(A_j) \right)^T \cdot \left( v(B_i) - \frac{1}{n} \sum_{j=1}^{n} v(B_j) \right) \right]\tag{2}$$

Where $u(\bullet)$ and $v(\bullet)$ represent the RFF mapping functions, n represents the number of input images.

We use $\mathbf{w} \in \mathbb{R}_+^n$ to represent the sample weights and $\sum_{i=1}^{n} w_i = n$ in the Weight Learner as shown in Fig. 3. After weighting, the partial cross-covariance matrix for variables A and B is as follows:

$$\Sigma_{AB;\mathbf{w}} = \frac{1}{n-1} \sum_{i=1}^{n} \left[ \left( w_i u(A_i) - \frac{1}{n} \sum_{j=1}^{n} w_j u(A_j) \right)^T \cdot \left( w_i v(B_i) - \frac{1}{n} \sum_{j=1}^{n} w_j v(B_j) \right) \right]\tag{3}$$

$A_i$ and $B_i$ $(i \in [1, n])$ are sampled from the feature distribution of $A$ and $B$.

The objective function of weight learner is:

$$\mathbf{w}^* = \arg \min_{\mathbf{w} \in \Delta_n} \sum_{1 \leq i \leq j \leq m} \|\Sigma_{AB;\mathbf{w}}\|_F^2\tag{4}$$

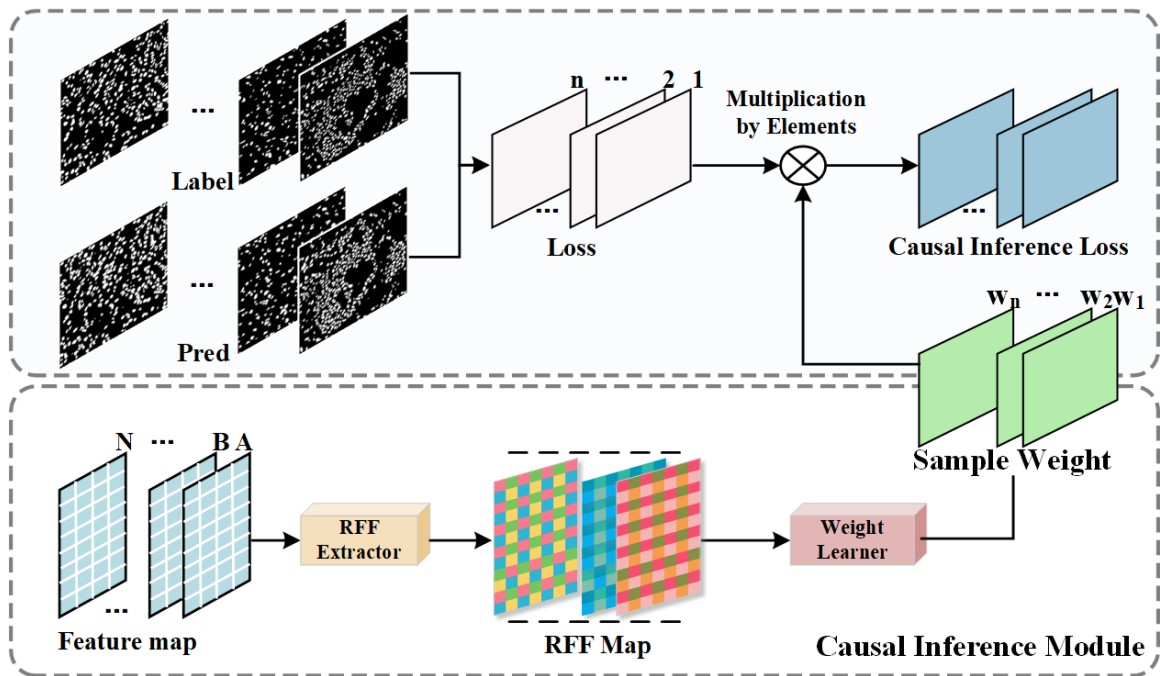

Figure 3: Detailed learning procedure of Causal Inference Module.

Where m represents the number of features and $\Delta_n = \{\mathbf{w} \in \mathbb{R}_+^n \mid \sum_{i=1}^n w_i = n\}$. Hence, weighting training samples with the optimal $\mathbf{w}^*$ can mitigate the dependence between features and consequently remove spurious correlations between features to the greatest extent.

Finally, the causal inference loss is as follows:

$$L_{CIM} = \sum_{i=1}^n w_i \otimes f(L_{CE}(X_i, y_i)) \qquad (5)$$

Where i represents the size of batch size. $L_{CE}(\cdot, \cdot)$ is the cross-entropy loss and $\otimes$ represents element-wise multiplication. $f(\bullet)$ represents the flattening operation.

Furthermore, we leverage dice loss and focal loss (Lin et al., 2017) to optimize our framework simultaneously. The dice loss is a common loss in the segmentation tasks. And the focal loss aims to solve the problem of category imbalance by reducing the impact of easily classifiable samples and enhancing the challenging ones:

$$L_{FL} = -\alpha_t (1 - p_t)^\gamma \log(p_t) \qquad (6)$$

Where $\alpha_t$ is a balancing coefficient for Nuclei and background. $p_t$ represents the predicted probability for the correct class. $\gamma$ is a tuning factor to adjust the weights of background. In this paper, we set $\alpha_t$ as 0.8 and $\gamma$ as 2.0.

Hence, the total loss is:

$$L = L_{CIM} + \lambda L_{Dice} + (1 - \lambda) L_{FL} \qquad (7)$$

Where $\lambda$ represents the coefficient of dice loss and $(1 - \lambda)$ represents the coefficient of focal loss. We set $\lambda$ as 0.5.

## 3 Experiments

### 3.1 Dataset and Experimental Details

The precise pixel-level labelling required for nuclei segmentation is a labour-intensive task. In order to evaluate the performance of the model, two smaller datasets were chosen: MoNuSeg (Kumar et al., 2017) and GLySAC (Doan et al., 2022). MoNuSeg includes tissue images from various tumour patients, which have been annotated by multiple healthcare providers. This results in discrepancies due to differences in the organs, patients and staining protocols. GLySAC covers a range of nuclei, including lymphocyte and cancer epithelial nuclei, with varied sample characteristics. The data was divided into three sets: training, validation, and test. The ratio of the training set to the validation and test sets was 6:2:2. The experiments were conducted using PyTorch 2.2.1 on an NVIDIA RTX4090 GPU. The model was trained with the AdamW optimiser and a cosine annealing learning rate strategy (Loshchilov and Hutter, 2016) (initial rate 1e-3). The batch size was 8, and the training was conducted over 400 epochs. Early stopping was used to prevent overfitting. The images were resized to $224 \times 224$ and augmented with a variety of techniques, including horizontal flipping, rotation, Gaussian blur, colour intensity enhancement, and random cropping.

### 3.2 Evaluation metrics

In this work, we use two core metrics to evaluate model performance: the Dice Coefficient (DSC) and the Mean Intersection over Union (mIoU). DSC is suitable for binary image segmentation and measures sample similarity, while mIoU assesses the overlap between predicted and actual areas. These metrics together provide a precise evaluation of performance, verifying the model's effectiveness in complex image segmentation tasks.

### 3.3 Experimental Results

A comparison of the performance of seven state-of-the-art (SOTA) models was conducted. As demonstrated in Table 1, our framework exhibits superior performance compared to Attention U-Net, achieving 3.59% higher mIoU and 2.61% higher DSC on the MoNuSeg dataset. It is noteworthy that the proposed approach outperforms transformer-based models such as TransUNet by 10.95% in mIoU and 8.16% in DSC. In comparison to UNet, our model demonstrates an improvement of 2.71% in mIoU and 2.04% in DSC on the GLySAC dataset. In comparison with TransUNet, the model demonstrates an increase of 11.46% in mIoU and 9.27% in DSC. These results demonstrate that CDNet attains the highest scores in the field of pathology image segmentation.

Nuclei segmentation is a crucial aspect of pathology image analysis. To demonstrate the superiority of CDNet, we conducted a comparative analysis with some of the more advanced models. As illustrated in Fig. 4, CDNet exhibited superior performance in terms of accuracy and efficiency, particularly in distinguishing closely neighbouring nuclei and identifying nucleus boundaries.

Table 1: Performance Comparison: CDNet vs SOTA on MoNuSeg and GLySAC. (*: P < 0.05)

| Method | MoNuSeg | | GLySAC | |
|---|---|---|---|---|
| | mIoU(%)↑ | DSC(%)↑ | mIoU(%)↑ | DSC(%)↑ |
| U-Net [1] | 63.97±5.41* | 77.91±4.01* | 61.04±7.20 | 75.56±5.79 |
| U-Net++ [2] | 63.96±6.57* | 77.83±5.07* | 60.89±7.08 | 75.45±5.72 |
| Attention U-Net [3] | 65.63±5.47 | 79.13±3.99 | 59.02±7.46* | 73.95±6.26 |
| TransUNet [4] | 58.27±3.42* | 73.58±2.71* | 52.29±7.57* | 68.33±7.06* |
| Swin-UNet [5] | 62.20±5.53* | 76.56±4.23* | 56.86±7.42* | 72.21±6.37* |
| Hover-Net [9] | 61.69±3.83* | 76.24±2.89* | 57.45±6.44* | 72.76±5.41* |
| UCTransNet [10] | 65.51±5.49 | 79.04±4.03 | 60.93±7.19 | 75.48±5.74 |
| Ours | **69.22±4.12** | **81.74±2.89** | **63.75±7.51** | **77.60±5.97** |

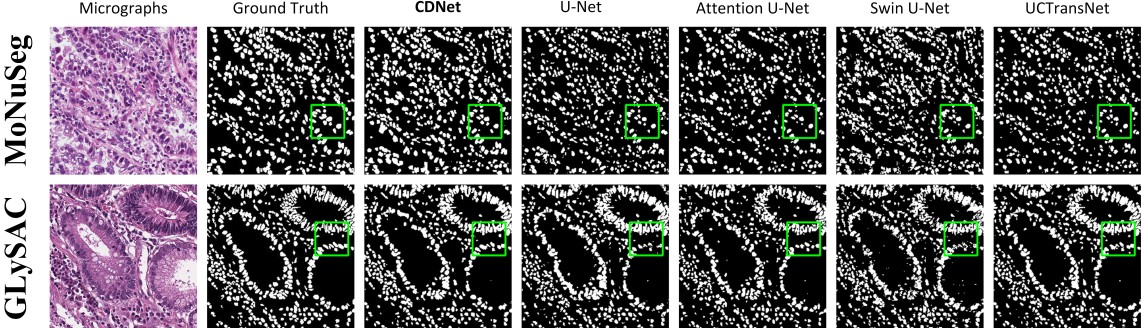

Figure 4: Comparison of the visualisation results of our framework with other SOTA methods on MoNuSeg and GLySAC.

## 3.4 Ablation Study

Ablation studies were conducted to evaluate the efficacy of key components in CDNet. As demonstrated in Table 2, the Backbone (comprising CNN downsampling and transformer) exhibited limited performance. The incorporation of DAC led to a 4.17% improvement in mIoU and a 3.02% improvement in DSC on MoNuSeg, as well as a 2.05% improvement in mIoU and a 1.58% improvement in DSC on GLySAC. This was achieved by enhancing edge recognition and feature fusion. The incorporation of CIM resulted in an improvement of 1.55% and 1.13% in mIoU and DSC, respectively, on MoNuSeg, and 0.91% and 0.54% on GLySAC, due to the elimination of spurious feature correlations. The combination of Backbone, DAC, and CIM led to an improvement in mIoU by 2.25% and DSC by 1.63% on MoNuSeg, and by 2.35% and 1.82% on GLySAC. These results demonstrate that CDNet enhances cell nucleus segmentation in pathology image analysis.

In addition, we show the visualisation of the Grad-CAM (Selvaraju et al., 2017) feature activation before and after applying these two modules, as shown in Fig. 5. DAC effec-

Table 2: Quantitative evaluation of the proposed crucial modules in CDNet.

| Module | | | MoNuSeg | | GLySAC | |
|---|---|---|---|---|---|---|
| Backbone | DAC | CIM | mIoU(%)↑ | Dice(%)↑ | mIoU(%)↑ | Dice(%)↑ |
| ✓ | | | 64.67±4.53 | 78.46±3.34 | 61.40±8.02 | 75.78±6.56 |
| ✓ | ✓ | | 68.84±4.01 | 81.48±2.82 | 63.45±7.67 | 77.36±6.09 |
| ✓ | | ✓ | 67.99±4.48 | 80.87±3.13 | 62.31±9.44 | 76.32±8.14 |
| ✓ | ✓ | ✓ | **69.22±4.12** | **81.74±2.89** | **63.75±7.51** | **77.60±5.97** |

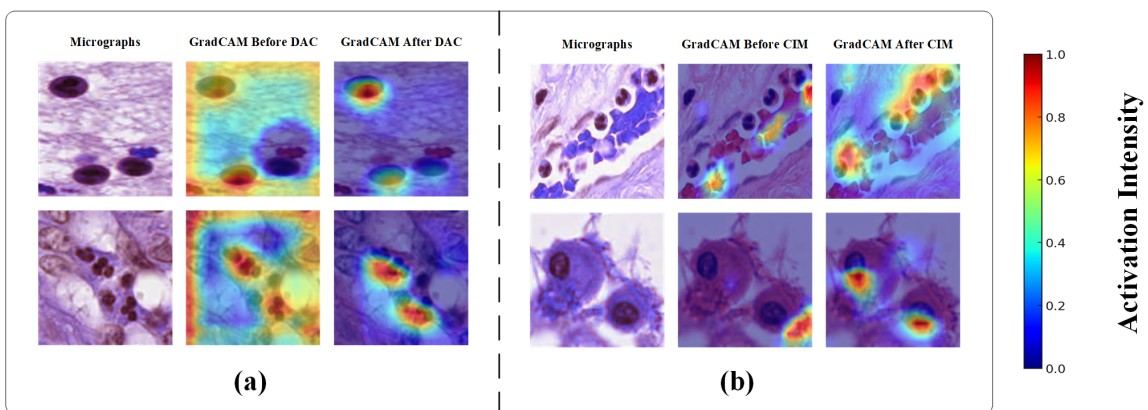

Figure 5: The activation of features was visualised using GradCAM. (a) The micrographs are displayed from left to right, with the GradCAM results with DAC enabled on the left and the results with-out DAC enabled on the right. (b) The images displayed from left to right are micrographs, GradCAM results with CIM enabled, and results without CIM enabled.

tively reduces the model's focus on background regions compared to the case without these modules, whereas CIM results in migrating the model's focus from irrelevant features to the nucleus.

## 4  Conclusion

In this paper, focusing on the challenges of domain shifts and blurred edges in the Nuclei segmentation tasks, we propose a novel CDNet that combines CIM with DAC. Compared to other methods, our CDNet can effectively improve the results of Nuclei segmentation by removing spurious correlations between features, and focusing more on the correlations between features and labels. Meanwhile, it improves the accuracy of Nuclei edges recognition by optimizing various types of semantic information for feature fusion. The extensive experiments demonstrate that our CDNet can effectively alleviate the issues in the Nuclei segmentation and achieves a robust performance.

## Acknowledgments and Disclosure of Funding

This work was supported in part by National Natural Science Foundation of China (Grant No. 62171130, 62172197, 62301160), the Natural Science Foundation of Fujian Province (Grant No.2020J01573, 2022J01131257, 2022J01607).

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
