# OpenReview forum: "CDNet: Causal Inference inspired Diversified Aggregation Convolution for Pathology Image Segmentation"
_MICCAI.org/2024/Workshop/COMPAYL — COMPAYL 2024_

### Official Review · Reviewer_UnFv · 2024-07-03
**Review for Dawei Fan**

**Custom Rating:** 4
**Confidence:** 1

**Review:**

Quality:
The quality of the work is overall very good.
Pros:
The detailed comparative analysis with SOTA models underscores the robustness of the proposed method and more important the model outperforms the compared SOTA models.

Clarity: The clarity of the work is overall okay.
Pros:
Clear and logical structure and organization of the work.
Cons:
The use of technical terminology, mainly in the methodology sections, may pose a challenge for readers not well-versed in the field.

Originality:
The work presents original contributions through the innovative combination of CIM and DAC incorporated in the CDNet framework, improving segmentation accuracy.

Significance:
The work has significant potential to improve pathology image segmentation

---

### Official Review · Reviewer_A4QJ · 2024-07-09
**Application of StableNet feature decorrelation plus attention equals better performance**

**Custom Rating:** 5
**Confidence:** 4

**Review:**

The application of StableNet's learned weights for decorrelation in nucleus segmentation makes a lot of sense. It's good to see the application of a method that directly addresses the problem where the model learns to correlate various types of background noise. The ablation does appear to show that both DAC and CIM are necessary to achieve such performance.

The question of whether DAC is a generally better way of integrating multiscale feature maps or not remains to be seen. Due to the lack of simpler baselines replacing DAC with other feature aggregators, it is difficult to say whether DAC is a superior feature aggregator.

Overall the paper proposed a strong method that makes sense. It demonstrated good performance.

---

### Official Review · Reviewer_Q5ZC · 2024-07-15
**Interesting method yet questionable application**

**Custom Rating:** 2
**Confidence:** 5

**Review:**

Summary of paper:

The authors develop a novel deep learning model by integrating two previously developed modules into a mixed convolutional and transformer model. These diversified aggregation convolution and causal inference module are then shown to improve dice and mean IoU for binary segmentation in two datasets. Included ablation study and some gradcam examples further demonstrate the impact of the introduced modules.

Strengths:

- The proposed method integrates two interesting modules and takes a different approach to the nuclei segmentation task.
- The manuscript is well written with included examples, nice figures and an ablation study to support their claims.

Weaknesses:

Major:

- The proposed method is applied to a binary segmentation task instead of an instance segmentation task which misses the point of nuclei segmentation completely. The authors should include an instance segmentation evaluation and comparison or address how this binary segmentation output is supposed to be used for further downstream tasks in histopathology.
- The authors compare their method against e.g. HoVer-Net which also does instance segmentation and classification which is an unfair comparison. They also do not explain where the results for these models come from and perhaps the *-symbol is supposed to address that, but I cannot find an explanation for the star.
- The authors propose a combination of new methods and include an ablation study. Yet they do this with a single split. Please include cross-validation experiments to further support the claims.
- Ground-truth annotations are mostly done manually by human pathologists. These annotations are by design slightly random, especially in blurry borders. The method is supposed to accurately match the human segmentation, yet the human segmentation might also be wrong. Can the authors include reasons as to why a more accurate nuclei boundary segmentation would be beneficial?

Minor:

- The images without overlay in Figure 5 have the wrong color which usually happens if you load images with opencv and forget to change channels from BGR to RGB.

Other comments:

- It would be nice to know how computationally expensive this approach is compared to the other included methods. How long does inference for a single tile take in comparison?

Code, model weights, and/or data availability:
- not available

Conclusion:

The authors propose an interesting segmentation method which includes two previously developed building blocks for deep learning models and evaluate the method on a histopathology task. Yet, the authors should include valid comparisons against models that solve the same task and include a thorough evaluation using cross-validation. Moreover, and more importantly, the authors need to explain what kind of problem they are trying to solve with this approach.

---

### Decision · Program_Chairs · 2024-07-16

Accept